# Endometriotic lesions exhibit distinct metabolic signature compared to paired eutopic endometrium at the single-cell level

Meruert Sarsenova [1,2,3,4,9], Ankita Lawarde[1,4,9], Amruta D. S. Pathare [1,4], Merli Saare[1,4],
Vijayachitra Modhukur[1,4], Pille Soplepmann[5], Anton Terasmaa[6], Tuuli Käämbre[6],
Kristina Gemzell-Danielsson [2,3], Parameswaran Grace Luther Lalitkumar[2,3],
Andres Salumets [1,4,7,8,10] ✉ & Maire Peters[1,4,10]

Current therapeutics of endometriosis focus on hormonal disruption of endometriotic lesions (ectopic endometrium, EcE). Recent findings show higher glycolysis utilization in EcE, suggesting non-hormonal strategy for disease treatment that addresses cellular metabolism. Identifying metabolically altered cell types in EcE is important for targeted metabolic drug therapy without affecting eutopic endometrium (EuE). Here, using single-cell RNA-sequencing, we examine twelve metabolic pathways in paired samples of EuE and EcE from women with confirmed endometriosis. We detect nine major cell types in both EuE and EcE. Metabolic pathways are most differentially regulated in perivascular, stromal, and endothelial cells, with the highest changes in AMPK signaling, HIF-1 signaling, glutathione metabolism, oxidative phosphorylation, and glycolysis. We identify transcriptomic co-activation of glycolytic and oxidative metabolism in perivascular and stromal cells of EcE, indicating a critical role of metabolic reprogramming in maintaining endometriotic lesion growth. Perivascular cells, involved in endometrial stroma repair and angiogenesis, may be potential targets for non-hormonal treatment of endometriosis.

Endometriosis is a chronic, steroid-dependent condition that affects approximately 10% of reproductive-aged women[1]. Endometriosis is characterized by the ectopic growth of endometrial-like tissue, which exerts inflammatory, angiogenic, proliferative and fibrotic processes[1–3]. In 80% of cases endometriotic lesions are localized superficially on the peritoneum[3]. To understand the alterations in cellular status that facilitate cell growth and survival at peritoneal sites, it is essential to closely examine the lesions and the paired endometrial samples from the same women affected by endometriosis at single-cell resolution. Recent single-cell RNA sequencing (scRNA-seq) studies revealed the heterogeneous cellular composition of endometrium, as well as endometriotic lesions, and their microenvironment[4–6].

Cellular proportions and cell-specific transcriptomic profiles of endometrium change across the menstrual cycle under the control of endocrine factors[7]. Steroid hormones define the phase of the menstrual cycle and lead to morphological and transcriptomic changes in endometrial cells that reflect underlying processes, such as cell proliferation and angiogenesis[8]. These processes require an increase in energy production and substrate biosynthesis[9]. Steroid hormones also activate cell cycle regulators[10–12] that in turn regulate metabolic reprogramming[13]. The ability

[1]Department of Obstetrics and Gynaecology, Institute of Clinical Medicine, University of Tartu, Tartu, Estonia. [2]Division of Neonatology, Obstetrics and Gynecology, and Reproductive Health, Department of Women's and Children's Health, Karolinska Institutet, Stockholm, Sweden. [3]WHO Collaborating Centre, Gynecology and Reproductive Medicine, Karolinska University Hospital, Stockholm, Sweden. [4]Competence Centre on Health Technologies, Tartu, Estonia. [5]Tartu University Hospital Women's Clinic, Tartu, Estonia. [6]Laboratory of Chemical Biology, National Institute of Chemical Physics and Biophysics, Tallinn, Estonia. [7]Division of Obstetrics and Gynaecology, Department of Clinical Science, Intervention and Technology (CLINTEC), Karolinska Institutet, Stockholm, Sweden. [8]Department of Gynecology and Reproductive Medicine, Karolinska University Hospital, Stockholm, Sweden. [9]These authors contributed equally: Meruert Sarsenova, Ankita Lawarde. [10]These authors jointly supervised this work: Andres Salumets, Maire Peters. ✉e-mail: andres.salumets@ki.se

of a cell to transit through the cell cycle in response to tissue environmental cues is also dependent on adequate nutrient supply and its metabolic activity and plasticity, referred as adaptive metabolic response[14].

The endometrium of healthy fertile women is finely controlled by ovarian hormones, and metabolic signaling pathways regulate cellular homeostasis[15]. However, hormonal imbalances, hypoxia and elevated oxidative stress, may affect endometrial tissue processes. A hypoxic microenvironment and altered levels of estradiol (E2) and progesterone (P4) have been observed in the ectopic endometrial tissue from women with endometriosis[16–18], and they can alter cellular metabolism via the regulation of metabolic pathways. Hypoxic microenvironment was also shown to affect metabolic activity in endometriosis[19]. Previous in vitro studies reported metabolic switch in endometriosis from oxidative to glycolytic metabolism[19–21]. This shift resembles the Warburg effect, a mechanism actively employed by cancer cells, where cells use aerobic glycolysis to divide and produce biomass and energy at a higher rate[22,23]. Several studies have discussed new strategies of nonhormonal endometriosis management by addressing cellular metabolism via glucose transporter GLUT4[24], or by targeting key enzymes regulating pyruvate utilization, like LDHA[25], PDH or PDK1[26].

In our study, we investigated different cell types and their metabolic networks in peritoneal lesions (ectopic endometrium, EcE) compared to eutopic endometrium (EuE) using scRNA-seq analysis. This knowledge may facilitate understanding of the metabolic plasticity of EcE in maintaining its homeostasis and growth in a context of unfavorable microenvironment.

## Results
The study was designed as depicted in Fig. 1A. Eight paired samples of EcE and EuE from four women with confirmed endometriosis (Fig. 1B) were examined and a total of 16,924 cells were obtained from the single-cell transcriptomic analysis. After quality control and removal of doublets, 7279 and 7538 cells from EuE and EcE, respectively, remained for further analysis. The average number of genes per cell was 2480 (range 1762–3199).

### EcE is enriched for perivascular, endothelial, and immune cells
The cells in EuE and EcE were annotated and initially grouped into 16 cell clusters according to the expression of cell type-specific marker genes (Supplementary Fig. 1), followed by re-clustering into nine major cell populations of stromal, endothelial, perivascular, immune, epithelial, cycling stromal cells, macrophages, B cells, and unknown cell population (Fig. 1C). Each cluster was determined based on the differentially expressed genes (DEGs) in that cluster versus overall expression in the rest of the clusters (Supplementary Fig. 2 and Fig. 1D). Uniform Manifold Approximation and Projection (UMAP) of individual samples (Supplementary Fig. 3) identified a similar cell clustering within EuE and EcE groups. The cycling stromal cells comprised a population of fibroblasts that expressed cell cycle genes and clustered separately from the main stromal cell population. The unknown cluster included the cells that did not express marker genes that could assign them to a particular cluster.

The statistical analysis of cell proportions between EcE and EuE revealed a significant difference in all nine cell populations, as shown on Fig. 1E and Supplementary Table 1. In EuE, the stromal, epithelial and cycling stromal cell clusters were larger compared to EcE (two-sided Fisher's exact test, $p < 1.13 \times 10^{-103}$, $p = 1.03 \times 10^{-93}$, and $p = 1.13 \times 10^{-103}$, respectively), while in EcE, the perivascular, endothelial, immune, macrophage, B cell and unknown cell populations were larger than in EuE (two-sided Fisher's exact test, $p < 1.13 \times 10^{-103}$, $p < 1.13 \times 10^{-103}$, $p = 5.56 \times 10^{-33}$, $p = 5.93 \times 10^{-50}$, $p = 7.39 \times 10^{-63}$ and $p = 2.62 \times 10^{-4}$, respectively). Among 9 cell clusters, stromal, perivascular, endothelial and immune clusters were the largest in size in both EuE and EcE (Fig. 1E). The epithelial cell cluster comprised a small cell population in both EuE and EcE, probably due to the cell loss during the tissue dissociation procedure.

### Metabolic reprogramming of perivascular, stromal and endothelial cell populations in EcE
To explore metabolic activity in EcE and EuE, we analyzed the gene expression of 12 metabolic pathways: two regulatory pathways for cellular

metabolism, proliferation and angiogenesis (AMP-activated protein kinase (AMPK) signaling, Hypoxia-Inducible Factor-1 (HIF-1) signaling), and 10 catabolic and/or anabolic metabolic pathways (pyruvate metabolism, oxidative phosphorylation (OXPHOS), pentose phosphate pathway, glycolysis/gluconeogenesis, citrate cycle (tricarboxylic acid cycle, TCA), fatty acid (FA) degradation, glutathione metabolism, alanine, aspartate and glutamate (Ala, Asp and Glu) metabolism, FA biosynthesis, and FA elongation).

Among nine cell types, perivascular, stromal, and endothelial cell populations had the largest proportions of statistically significant DEGs calculated to the total number of genes in each pathway in the comparison of EcE vs EuE (Fig. 2A, Supplementary Table 2). Immune, epithelial, cycling stromal and unknown cell types had a small number of metabolic DEGs, while macrophages and B cells exhibited no significant difference between EcE and EuE. In perivascular cells, FA elongation (12/27, 44.4%), FA degradation (18/43, 41.9%), pentose phosphate pathway (11/30, 36.7%), HIF-1 signaling (39/109, 35.8%), and pyruvate metabolism (16/47, 34.0%), exhibited the highest percentages of DEGs. In stromal cell population, AMPK signaling (35/121, 28.9%), FA elongation (7/27, 25.9%), pyruvate metabolism (12/47, 25.5%), FA degradation (10/43, 23.3%), and glutathione metabolism (13/57, 22.8%) were among the top altered pathways. While in endothelial cells, glutathione metabolism (12/57, 21.1%), AMPK signaling (23/121, 19%), HIF-1 signaling (21/109, 19.3%), Ala, Asp and Glu metabolism (7/30, 18.9%), and FA biosynthesis (3/18, 16.7%) had the highest proportions of DEGs.

Next, we applied single-cell pathway analysis (SCPA) to evaluate the activity of 12 metabolic pathways in all cell types together. According to this analysis, a higher Q value refers to a higher difference in pathways between EcE and EuE, considered as two conditions. The analysis ranked HIF-1 and AMPK signaling pathways at the top of the list with Q values 6.2 and 6.0, respectively (Fig. 2B). We also compared the activity of metabolic pathways in each cell type individually. Similarly, HIF-1 and AMPK signaling pathways were top-ranked across all cell types (Supplementary Table 3). Perivascular, stromal, and endothelial cells showed the highest Q values for most of the pathways compared to other cell types (Fig. 2C), indicating high pathway differences between EcE and EuE.

For further analyses, we focused on three populations of endothelial, perivascular, and stromal cells that exhibited the highest differences in the regulation of metabolic pathways based on gene expression and SCPA analyses between EcE and EuE. We examined the contribution of each patient to the results of metabolic pathway activity in these three cell types using GSVA analysis. The results demonstrated that the activity patterns of all metabolic pathways were similar between individuals in the three cell types (Supplementary Fig. 4). Additionally, we performed the analysis of transcription factors (TF) associated with metabolic genes (Supplementary Data 1). Among the top 25 TFs (Fig. 2D), *BRCA1* was most active in perivascular cells of EcE and endothelial cells of EuE and EcE. We found an activation of TFs related to proliferation and survival (*ATF1* in endothelial cells of EcE, *MECOM* in perivascular cells of EcE, *ETS2*, *NFYA* and *XBP1* in stromal and perivascular of EuE), associated with epithelial–mesenchymal transition (*DNMT1*, *EZH2*, *HNF1B*, *MYB* in stromal cells of EuE), and cell adhesion, angiogenesis and inflammation (*NR2F2* in perivascular cells of EcE, *NCOR2* in stromal cells of EuE), as shown on Fig. 2D.

Next, we checked the percentages of statistically significant up- and down-regulated DEGs in perivascular, stromal and endothelial cell types across 12 metabolic pathways (Table 1). The analysis was performed by comparing EcE to EuE, and the reported statistically significant DEGs are given accordingly (Supplementary Data 2).

Metabolic pathways are interconnected, and some enzymes with multiple functions play a role in different but related processes (Fig. 2E). As a result, certain genes encoding these enzymes are present in several pathways as discussed below. The DEGs (and their corresponding $\log_2$fold change, $\log_2$FC) of metabolic pathways in perivascular, endothelial, and stromal cell types of ectopic endometrium compared to eutopic endometrium are presented in Table 2. We grouped the results of differential gene expression

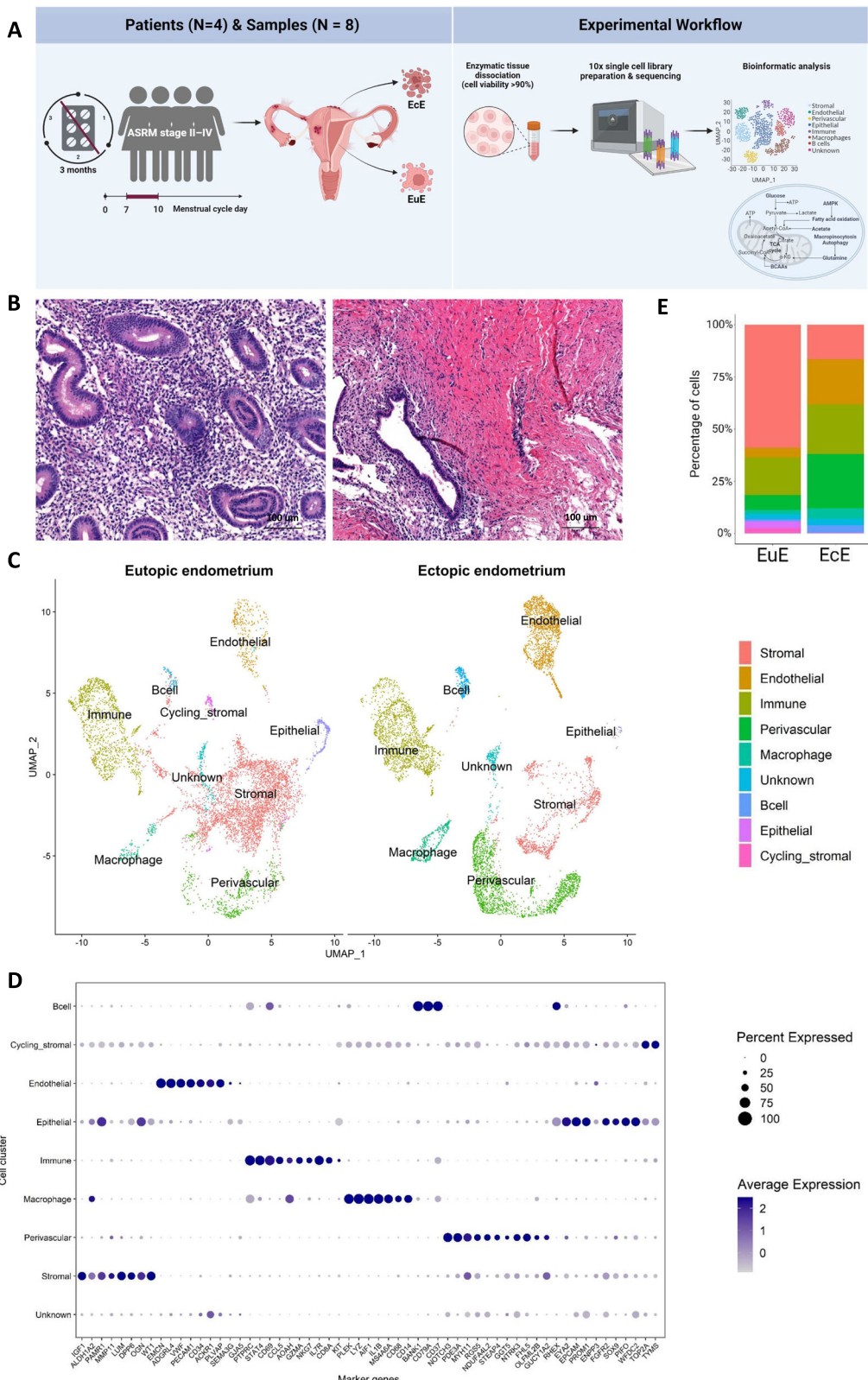

of metabolic pathways by the relevance of the pathways to the regulation or type of energy metabolism (regulatory, glycolytic, oxidative, and biosynthetic). Representative boxplots of DEGs from each group, visualizing the range of the level of expression across the samples, are presented in Supplementary Figs. 5–7.

## Regulatory metabolic pathways (AMPK signaling pathway and HIF-1 signaling pathway)

AMPK and HIF-1 signaling pathways exhibited a differential regulation of several genes involved in cellular processes like metabolism, proliferation, and angiogenesis. We found a transcriptomic activation of AMPK via HIF-1

**Fig. 1 | An overview of the single cell landscape of peritoneal lesions and eutopic endometrium. A** Illustration of the study design. Paired ectopic endometrium (EcE, $N = 4$) and eutopic endometrium (EuE, $N = 4$) tissue samples from four women with endometriosis during the proliferative phase of menstrual cycle were enzymatically dissociated. The isolated cells were subject to 10x Genomics single cell library preparation and RNA sequencing. Further bioinformatic analyses of cell populations and metabolic changes were applied. Created with BioRender.com. **B** Representative images of haematoxylin and eosin staining of EuE and EcE from a woman with endometriosis, showing morphological differences between paired endometrial tissues. Scale bar 100 μm. **C** Uniform Manifold Approximation and Projection

(UMAP) plot of nine major cell clusters. The cycling stromal cell cluster in EcE was represented by few cells and due to the low cell count and wide distribution of the dots representing this cluster, the cluster name is omitted from the figure to avoid confusion with the cluster localization. **D** Expression of marker genes in nine major cell types, the size of the dots shows the percentage of cells expressing the genes within the cell population, and the color of the dot corresponds to the average gene expression. Each cell type exhibited the expression of cell-type representative marker genes. **E** Bar plots representing the relative ratios of 9 major cell populations in EcE and EuE. Perivascular, endothelial, and immune cell populations were larger in EcE compared to EuE, while stromal cell population was larger in EuE.

target genes, *CAMK2A*, *CAMK2D* and *CAMK2G*, upregulated in all three cell types of EcE (Table 2). Apart from that, perivascular and stromal cells in EcE had a high expression of AMPK activator *CAB39L* and a low expression of AMPK inhibitor *PP2A*. *PRKAA1*, *PRKAA2* and *PRKAG2* genes encoding subunits of AMPK were overexpressed in all three populations of EcE.

HIF-1 and AMPK-regulated genes from energy-producing pathways that utilize glucose and FAs were overexpressed in EcE. In perivascular and stromal cell types, we found an upregulation of *TBC1D1* and *GLUT4* genes involved in glucose uptake. However, glycolysis activators *PFKFB3* (in perivascular cells) and *PFKFB4* (in perivascular, stromal, and endothelial cells) were downregulated in EcE. *CD36* (in perivascular cells), *CPT1A* (in perivascular and stromal cells), and *MLYCD* (in perivascular cells), involved in FA uptake, transport to mitochondria and β-oxidation were upregulated in EcE. In addition, in all three clusters of EcE, there was an overexpression of *ACACB* gene converting acetyl-CoA to malonyl-CoA that can be further used for FA synthesis. At the same time, an inhibitor of oxidative metabolism *PDK1*, was upregulated in perivascular cells of EcE.

In EcE, we found both up- and downregulation of genes associated with biosynthetic metabolism. For example, genes involved in FA and steroid biosynthesis, such as *ACACB* (upregulated in all three cell types), *SREBF1* (downregulated in perivascular, stromal, and endothelial cells), *LIPE* and *SCD* (downregulated in stromal cells). Key regulators and activators of gluconeogenesis (*CREB5* in perivascular, stromal, and endothelial cells), glycogen synthesis (*GYS1* in stromal cells), and sterol synthesis (*HMGCR* in perivascular cells) were inhibited in EcE. *RPTOR*, involved in the activation of protein synthesis was downregulated in perivascular and stromal cells, while *EEF2K* was overexpressed in all three clusters of EcE.

Additionally, we observed HIF-1 and AMPK-mediated effect on cell cycle progression, proliferation, and cell survival. *CDKN1A* (in perivascular and stromal cells) and *CDKN1B* (in perivascular, stromal cell and endothelial cells) were upregulated in EcE. *CCNA2* was inhibited in perivascular and endothelial cells of EcE, while *CCND1* was upregulated in perivascular and stromal cells but downregulated in endothelial cells. The anti-apoptotic gene *BCL2* was upregulated in stromal cells and is downregulated in endothelial cells of EcE. In stromal cells, we also observed a downregulation of autophagy mediator *ULK1*.

We found a differential expression of HIF-1-mediated genes involved in angiogenesis in EcE compared to EuE. *VEGFA* (in perivascular cells), *ANGPT1* (in perivascular and stromal cells), and *ANGPT4* (in perivascular cells) were upregulated, while *ANGPT2* (in perivascular and endothelial cells), known ANGP1 antagonist, and *TEK* (in perivascular cells) were downregulated. Additionally, there was an overexpression of VEGF activator *IL6*, its receptor *IL6R* (both in perivascular and endothelial cells), and *FLT1* (in stromal cells).

### Glycolytic metabolism

In EcE, we found an overexpression of key glycolysis-related genes like *ALDOA*, *ALDOC* and *PGAM2* (in perivascular cells), *ENO2* (in perivascular, stromal and endothelial cells), and *PFKP* (in perivascular and stromal cells). Interestingly, we observed a downregulation of *HK1* (in perivascular and stromal cells), and *HK2* (in perivascular cells) but an upregulation of *GCK* known to take the role of hexokinases and maintain glycolysis.

In perivascular and endothelial cells of EcE, we found a differential regulation of genes related to glucogenic amino acid metabolism with some

genes upregulated, (for example *ASPA* in perivascular cells), and other genes downregulated, like *GPT2* in perivascular and endothelial cells. In stromal cells the gene expression of this pathway was similar in EcE and EuE. We also observed an overexpression of lactate dehydrogenases (LDH) converting pyruvate to L- and D-lactate, namely *LDHA* in perivascular cells, and *LDHA* and *LDHD* in stromal cells of EcE. In endothelial cells *LDHC* was upregulated, while *LDHB* was downregulated.

### Oxidative metabolism

In EcE, we identified an upregulation of genes related to the formation of acetyl-CoA from FAs, called β-oxidation, such as *ACADL*, *CPT1A*, and *CYP2U1* in perivascular and stromal cells, and *ACADS* in endothelial cells. *PDHA*, *PDHB* and *DLD* genes involved in the conversion of pyruvate to acetyl-CoA, were also upregulated in perivascular and stromal cells of EcE. In endothelial cells, we found a downregulation of *ME1* that converts malate (TCA cycle intermediate) to pyruvate. Some key genes of TCA cycle like *IDH3B*, *IDH2*, *PCK1*, *SDHC*, *SUCLA2*, *SUCLG1* and *SDHD* were upregulated in EcE.

In ectopic perivascular cells, 24% of genes, encoding five multiprotein complexes of the electron transport chain of OXPHOS were overexpressed. Stromal cells exhibited an upregulation of *COX17*, *NDUFA4L2*, *NDUFB9*, *SDHD*, and *UQCRFS1* and a downregulation of *NDUFA6*, *NDUFB11*, *COX5A*, *COX6C*, and *UQCRQ*. In endothelial cells, *ATP6V1G2*, *COX7A2L*, and *SDHD*, were upregulated in EcE, while *LHPP* and *UQCRC1* were higher expressed in EuE.

Some genes of glutathione synthesis like *GPX3* (in perivascular and stromal cells) and *MGST3* (in perivascular, stromal cell and endothelial cells) were higher expressed in EcE, while other genes like *GPX7* (in ectopic perivascular, stromal cell and endothelial cells) and *PGD* (in ectopic perivascular, stromal cell and endothelial cells) were overexpressed in EuE.

### Biosynthetic metabolism

In EcE, we identified an overexpression of *OLAH* (in perivascular cells) that participates in FA release from FA synthase. Some genes, involved in the conversion of FA to acyl-CoA, were downregulated in perivascular cells (*ACSBG1*) and upregulated in endothelial cells (*ACSL5*) of EcE. Some genes from FA elongation pathway were up- or downregulated in perivascular cells (*ELOVL2*, *ELOVL4*, *ELOVL6*, *HACD1*, *HACD3*, *HACD4*, *HSD17B12* and *TECR*), and in stromal cells (*ELOVL2*, *ELOVL4*, *ELOVL6*, *HACD1*, and *HSD17B12*), and upregulated in endothelial cells (*ELOVL2* and *ELOVL7*).

In addition, in EcE, we observed an upregulation of genes involved in ribose and nucleotide synthesis and metabolism, like *PGM2* in perivascular cells, *PRPS1* and *TKTL1* in perivascular and stromal cells, and *RBKS* in all three cell types. However, *PGD* (in perivascular, stromal and endothelial cells), *PRPS2* and *PGLS* (in perivascular cells) genes, involved in oxidative steps of the pathway, were downregulated in EcE.

### Metabolic activity is similar in EuE from women with endometriosis vs controls

We examined whether metabolic activity differs between EuE from women with endometriosis ($N = 3$) and EuE from women without endometriosis (controls, $N = 3$) in the proliferative phase of menstrual cycle using external dataset[27]. First, we identified the same 9 major clusters in both sample groups (Supplementary Figs. 8 and 9). The metabolic

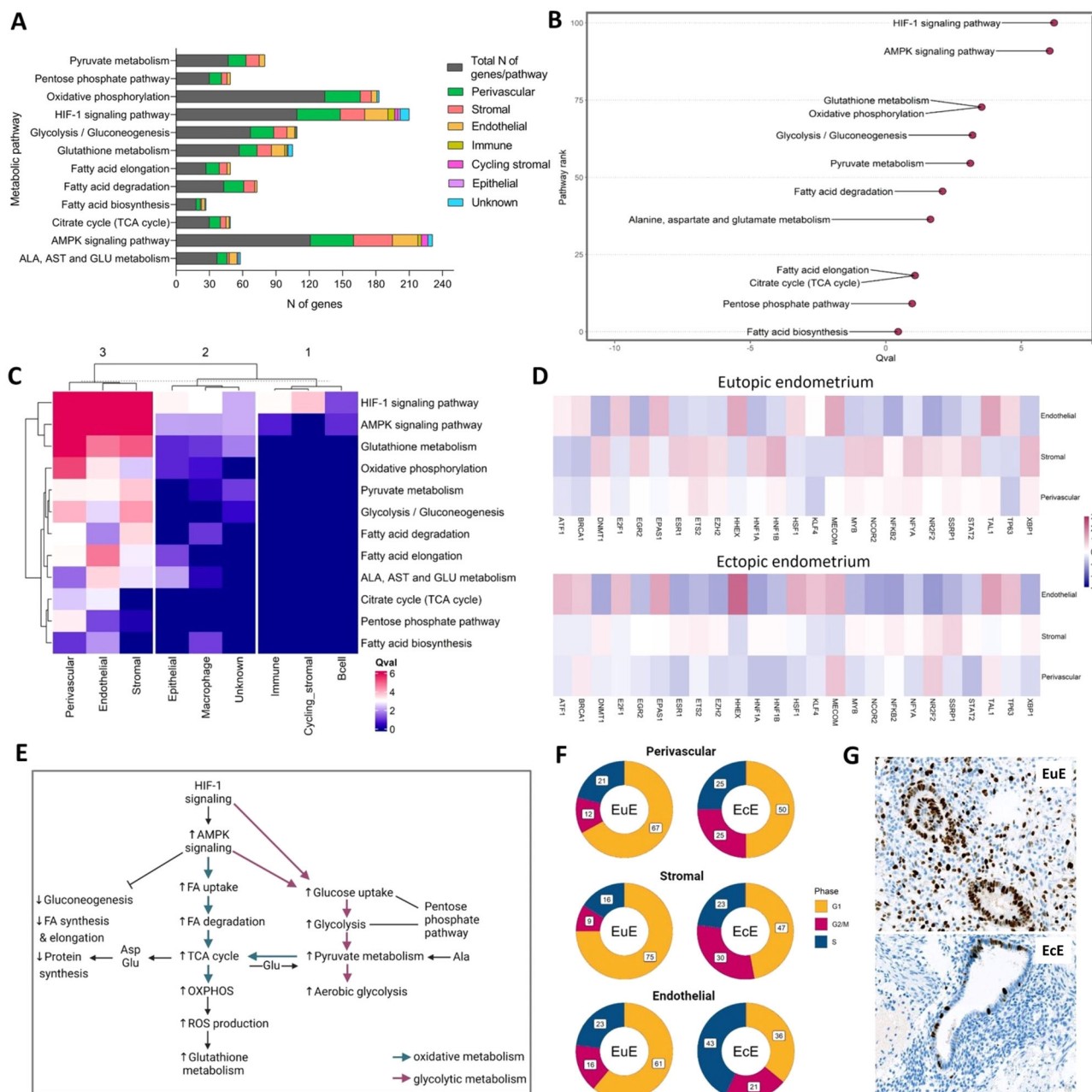

**Fig. 2 | Perivascular, stromal and endothelial cell populations of ectopic endometrium exhibited the activation of energy metabolic pathways and different proliferative activity compared to eutopic endometrium. A** A bar plot of the numbers of DEGs in 12 metabolic pathways across 9 major cell types (color-coded) and the total number of genes in a given pathway (in dark gray). **B** Ranking of metabolic pathways according to Q values (Qval) in merged cell populations of ectopic endometrium (EcE) compared to eutopic endometrium (EuE). Q value refers to a pathway difference between EcE and EuE, considered as two conditions. A higher number on y axis corresponds to a higher rank of the pathway. **C** A heatmap showing the Q values of 12 metabolic pathways across 9 cell types of EcE compared to EuE. The lower Q value is depicted in blue, the higher Q value in red. **D** The heatmap showing the activity of the top 25 transcription factors (TF) associated with metabolic genes in perivascular, stromal and endothelial cell populations in each

sample group. The score corresponds to the TF activity. **E** Interconnection between metabolic pathways involved in cellular metabolism. HIF-1 signaling and AMPK signaling are regulatory pathways for the metabolic pathways involved in glycolytic and oxidative metabolism, biosynthesis of macromolecules and nucleic acids. Pink arrows represent glycolytic metabolism, blue arrows represent oxidative metabolism. **F** The proportions of perivascular, stromal and endothelial cells in cell cycle phases in EuE and EcE. G1 – cell growth phase (in yellow), S – DNA synthesis phase (in blue), and G2/M - checkpoint and mitosis phase (in red). The numbers in white squares correspond to the percentages of the cells in a given cell cycle phase. **G** The representative images of immunostaining for Ki67 proliferation marker of paired EuE and EcE (peritoneal lesion) at the proliferative phase of menstrual cycle from a woman with endometriosis. Scale bar 200 μm.

pathway analysis of stromal, perivascular and endothelial cell populations revealed only few DEGs between EuE from women with endometriosis vs controls (Supplementary Table 4), thus indicating no major difference in the activity of metabolic pathways between EuE from women with endometriosis vs controls.

## Progesterone resistance and elevated estradiol signaling genes in EcE

As cellular metabolism in endometrium is partially regulated by steroid hormones, we next examined the expression levels of steroidogenic genes between EcE and EuE. The differential expression analysis revealed a lower

**Table 1 | Numbers of up- and down-regulated differentially expressed genes of metabolic pathways in perivascular, endothelial, and stromal cell types of ectopic endometrium compared to eutopic endometrium**

| Metabolic pathways | Total N of genes in a pathway | Cell type, N (%)[a] of upregulated (UP) or downregulated (DOWN) genes | | | | | |
| | | Perivascular | | Stromal | | Endothelial | |
| | | UP, N (%) | DOWN, N (%) | UP, N (%) | DOWN, N (%) | UP, N (%) | DOWN, N (%) |
|---|---|---|---|---|---|---|---|
| Ala, Asp and Glu metabolism | 37 | 5 (13.5%) | 4 (10.8%) | 1 (2.7%) | 1 (2.7%) | 4 (10.8%) | 3 (8.1%) |
| AMPK signaling pathway | 121 | 21 (17.4%) | 18 (14.9%) | 18 (14.9%) | 17 (14.0%) | 11 (9.1%) | 12 (9.9%) |
| Citrate cycle (TCA cycle) | 30 | 9 (30%) | 1 (3.3%) | 3 (10.0%) | 2 (6.7%) | 1 (3.3%) | 2 (6.7%) |
| Fatty acid biosynthesis | 18 | 3 (16.7%) | 1 (5.6%) | 1 (5.6%) | 0 | 2 (11.1%) | 1 (5.6%) |
| Fatty acid degradation | 43 | 15 (34.9%) | 3 (7%) | 10 (23.3%) | 0 | 2 (4,7%) | 0 |
| Fatty acid elongation | 27 | 6 (22.2%) | 6 (22.2%) | 4 (14.8%) | 3 (11.1%) | 2 (7.4%) | 1 (3.7%) |
| Glutathione metabolism | 57 | 7 (12.3%) | 9 (15.8%) | 8 (14%) | 5 (8,8%) | 4 (7%) | 8 (14%) |
| Glycolysis/ Gluconeogenesis | 67 | 19 (28.4%) | 2 (3%) | 10 (15%) | 2 (3%) | 4 (6.0%) | 3 (4.5%) |
| HIF-1 signaling pathway | 109 | 20 (18.3%) | 19 (17.4%) | 13 (11.9%) | 9 (8.3%) | 13 (11.9%) | 8 (7.3%) |
| Oxidative phosphorylation | 134 | 32 (23.9%) | 0 | 5 (3.7%) | 5 (3.7%) | 3 (2.2%) | 2 (1.5%) |
| Pentose phosphate pathway | 30 | 8 (26.7%) | 3 (10.0%) | 4 (13.3%) | 1 (3.3%) | 1 (3.3%) | 2 (6.7%) |
| Pyruvate metabolism | 47 | 16 (34%) | 0 | 12 (25.5%) | 0 | 3 (6.4%) | 2 (4.3%) |

[a]The percentages of the genes are calculated to the total N of genes in a given pathway.

expression of P4 receptor *PGR* in perivascular and endothelial cells of EcE (Table 3). *ESR2* (encoding estrogen receptor beta, ERβ) and *HSD17B8* genes involved in the activation of estrogen target genes and interconversion of a less potent estrogen E1 (estrone) and E2, respectively, were upregulated in ectopic perivascular cells. The expression level of *ESR1* (encoding estrogen receptor alpha, ERα) and *HSD17B2* involved in E2 conversion to E1 as well as the formation of an active metabolite of P4 were decreased in EcE compared to EuE. A downregulation of *SRD5A3, SRD5A1* and *AR* genes related to androgen receptor and androgen production, as well as *CYP11A1* and *HSD11B2* genes related to the synthesis and metabolism of steroids and cortisol, respectively, was observed in perivascular cell population in EcE. Stromal cells of EcE overexpressed *AKR1C1* and *AKR1C2* genes involved in the metabolism of P4, *HSD11B1* gene that encodes an enzyme producing cortisol, and *HSD17B6* and *HSD17B11* that enhance androgen metabolism. Endothelial cells also exhibited the upregulation of genes related to metabolism of P4 (*AKR1C1*) and ketones (*HSD17B11*). Together, the results of steroidogenesis analysis showed transcriptomic alterations of P4 signaling and increased P4 resistance (*PGR, AKR1C1, AKR1C2*), and elevated E2 activity via the overexpression of *ESR2* (ERβ), reduced conversion of E2 to E1 (*HSD17B2*), and the increased expression of *HSD17B8* involved in the synthesis of E2. At the same time, *ESR1* (ERα) expression was decreased in all three cell types, that may reduce the E2 effects. As the comparison of endometrial cell populations from endometriosis and control women (external dataset[27]) did not identify any statistically significantly expressed steroidogenic genes, the differences observed in our study reflect the changes in EcE.

**Differences in expression of cell cycle phase-specific genes between EuE and EcE**

As we observed changes in metabolic activity and steroidogenesis in perivascular, stromal and endothelial cell populations of EcE, we examined proportions of these three cell types by cell cycle phases in EcE and EuE. (Supplementary Table 5). In the perivascular, stromal, and endothelial cell populations of EuE, over 60% of the cells were in G1 phase. 16–23% of the cells expressed genes representative of S phase, while a lower proportion of cells with the range from 9% to 16% exhibited expression of genes related to the G2/M phase (Fig. 2F). In contrast, in EcE, a lower percentage of cells were detected to be in G1 phase (ranging from 36% to 50%), while higher proportions of cells expressed genes characteristic of the G2/M phase (21–30%) and S phase (23–43%) compared to EuE. The differences between ectopic and eutopic cell proportions were statistically significant

for each cell type in the three phases, according to Fisher's exact test (Supplementary Table 5), except for stromal cells in the G2/M phase.

**Differential expression and pathway analyses revealed altered proliferation, apoptosis, migration and angiogenesis processes in EcE**

We checked the overall differential gene expression profile and associated pathways in perivascular, stromal and endothelial cell populations. The differential expression analysis between EcE and EuE identified 5602, 3733 and 3233 DEGs (both up- and down-regulated) in perivascular, stromal, and endothelial cell types, respectively. We performed Kyoto Encyclopedia of Genes and Genomes (KEGG) pathway analysis and found the enrichment of extracellular matrix–receptor interaction, focal adhesion, PI3K-Akt signaling pathway, cell cycle, proteoglycans in cancer, cell adhesion molecules, AMPK and Hippo signaling pathways, and ribosome pathway to be altered between EcE and EuE tissues (Supplementary Figs. 10 and 11). These pathways are involved in cell–extracellular matrix interactions, cellular metabolism, cell survival, migration, angiogenesis, and cell proliferation. As Ki67 is one of the known markers of proliferation and it was higher expressed in perivascular and stromal cells of EuE, we decided to confirm its expression in tissue samples. The immunostaining (Fig. 2G, Supplementary Fig. 12) revealed that most of the glandular epithelial cells and some cells in the stroma of EuE were expressing Ki67, while few cells in the glands and stroma of EcE were positively stained.

Among the top 20 DEGs (Supplementary Fig. 13), we identified genes participating in the processes previously reported to play a role in the pathogenesis of endometriosis and other conditions characterized by an altered proliferation and invasive growth. In the perivascular cell compartment of EcE, we identified a downregulation of tumor suppressor *PAMR1*, metalloproteinase *ADAM12*, lncRNA *MEG8, DIO2* and *VCAN* genes, and an upregulation of antiapoptotic gene *NTRK2*. *SKAP2* was overexpressed in both perivascular and endothelial cell types of EcE. *ADIRF*, tumor-associated genes *CLU, FAM13C*, as well as *NGF, PRELP, RERG, PTGIS, GPC3* genes were among the top 20 overexpressed genes in EcE stromal cell population. *AQP1* was found to be upregulated in the endothelial and stromal cells of EcE compared with EuE. Gene ontology analysis identified cell cycle, cell adhesion, actin and integrin binding as the most enriched biological processes and molecular function in perivascular cells (Supplementary Fig. 14). Among stromal cells, the top enriched processes and functions included cell junction assembly, connective tissue development, transcription activator and regulator activity, and macromolecule

**Table 2 | Differentially expressed genes of metabolic pathways in perivascular, endothelial, and stromal cell types of ectopic endometrium compared with eutopic endometrium**

| Pathway group | Gene expression in EcE | DEGs[a] (log₂FC) | | |
|---|---|---|---|---|
| | | Perivascular cells | Stromal cells | Endothelial cells |
| Regulatory pathways | Upregulated | *CAMK2G* (1.74), *PDK1* (1.03), ***CDKN1A*** (3.81), ***CDKN1B*** (1.23), *VEGFA* (0.86), ***ANGPT1*** (3.89), *ANGPT4* (7.42), ***IL6*** (6.77), ***IL6R*** (3.09), *CAB39L* (1.90), *PRKAA1* (0.51), ***PRKAA2*** (3.24), ***PRKAG2*** (2.47), ***TBC1D1*** (2.19), *GLUT4* (3.08), *CD36* (4.48), ***CPT1A*** (1.51), *MLYCD* (0.95), ***ACACB*** (0.93), ***EEF2K*** (2.24), ***CCND1*** (1.65) | *CAMK2A* (1.58), ***CDKN1A*** (1.55), ***CDKN1B*** (0.86), ***BCL2*** (1.07), ***ANGPT1*** (1.87), *FLT1 (VEGFR-1,* 2.45), ***CAB39L*** (0.85), ***PRKAA2*** (1.24), ***PRKAG2*** (1.38), ***TBC1D1*** (1.86), ***GLUT4*** (3.16), ***CPT1A*** (1.07), ***ACACB*** (0.89), ***EEF2K*** (2.36), ***CCND1*** (2.36) | *CAMK2D* (1.28), ***CDKN1B*** (1.39), ***IL6*** (3.97), ***IL6R*** (1.92), ***PRKAG2*** (2.04), ***ACACB*** (2.47), ***EEF2K*** (0.68), ***CCND1*** (−0.93) |
| | Downregulated | ***ANGPT2*** (−0.66), *TEK* (−1.88), ***PP2A*** (−0.66), *PFKFB3* (−1.33), ***PFKFB4*** (−2.84), ***SREBF1*** (−1.20), ***CREB5*** (−1.9), *HMGCR* (−0.77), ***RPTOR*** (−0.41), ***EIF4EBP1*** (−0.82), ***CCNA2*** (−3.65) | ***PP2A*** (−0.97), ***PFKFB4*** (−1.90), ***SREBF1*** (−0.86), *LIPE* (−1.04), *SCD* (−1.51), ***CREB5*** (−1.53), *GYS1* (−0.58), ***RPTOR*** (−0.59), ***EIF4EBP1*** (−0.90), *ULK1* (−1.18) | ***BCL2*** (−0.98), ***ANGPT2*** (−1.45), ***PFKFB4*** (−1.59), ***SREBF1*** (−1.97), ***CREB5*** (−1.77), ***CCNA2*** (−1.90) |
| Glycolytic metabolism | Upregulated | *ALDOA* (0.89), *ALDOC* (1.71), ***ENO2*** (1.64), ***PFKP*** (0.91), *PGAM2* (6.17), *GCK* (1.64), *ASPA* (2.68), ***LDHA*** (0.99) | ***ENO2*** (2.60), ***PFKP*** (1.15), ***LDHA*** (1.39), *LDHD* (3.30) | ***ENO2*** (1.54), *LDHC* (2.83) |
| | Downregulated | ***HK1*** (−0.68), *HK2* (−2.45), ***GPT2*** (−1.48) | ***HK1*** (−0.84) | ***GPT2*** (−1.65), *LDHB* (−0.64) |
| Oxidative metabolism | Upregulated | *PDHA1* (0.67), *PDHB* (0.63), ***DLD*** (0.50), ***ACADL*** (7.76), ***CPT1A*** (1.51), ***CYP2U1*** (1.06), *IDH3B* (0.65), *PCK1* (1.86), *SDHC* (0.49), ***SDHD*** (0.77), *SUCLA2* (0.48), ***SUCLG1*** (0.52), ***GPX3*** (5.67), ***MGST3*** (1.96) | ***DLD*** (0.57), ***ACADL*** (4.98), ***CPT1A*** (1.07), ***CYP2U1*** (1.61), ***SDHD*** (0.64), ***SUCLG1*** (0.48), *COX17* (0.92), *NDUFA4L2* (3.67), *NDUFB9* (0.64), *UQCRFS1* (0.43), ***GPX3*** (3.77), ***MGST3*** (1.26) | *ACADS* (0.88), ***SDHD*** (0.57), *ATP6V1G2* (2.67), *COX7A2L* (0.57), ***MGST3*** (0.52) |
| | Downregulated | ***IDH2*** (−1.21), ***GPX7*** (−2.12), ***PGD*** (−1.52) | ***IDH2*** (−0.70), *NDUFA6* (−0.56), *NDUFB11* (−0.58), *COX5A* (−0.51), *COX6C* (−0.69), *UQCRQ* (−0.55), ***GPX7*** (−1.10), ***PGD*** (−0.89) | *ME1* (−1.63), ***IDH2*** (−1.81), *LHPP* (−0.79), *UQCRC1* (−0.65), ***GPX7*** (−1.13), ***PGD*** (−0.92) |
| Biosynthetic pathways | Upregulated | ***ACACB*** (0.93), *OLAH* (3.90), ***HACD1*** (2.86), *HACD4* (1.51), ***HSD17B12*** (1.29), *TECR* (0.59), *PGM2* (1.42), ***PRPS1*** (1.26), ***RBKS*** (1.99), *TKTL1* (2.80) | ***ACACB*** (0.89), ***ELOVL2*** (3.38), ***HACD1*** (2.30), ***HSD17B12*** (0.79), ***PRPS1*** (0.62), ***RBKS*** (0.65), *TKTL1* (1.87) | ***ACACB*** (2.47), *ACSL5* (0.69), ***ELOVL2*** (4.09), *ELOVL7* (7.36), ***RBKS*** (1.63) |
| | Downregulated | *ACSBG1* (−2.77), *ELOVL2* (−2.95), ***ELOVL4*** (−1.31), ***ELOVL6*** (−0.79), *HACD3* (−0.68), ***PGD*** (−1.52), *PRPS2* (−0.87), *PGLS* (−0.42) | ***ELOVL4*** (−0.94), ***ELOVL6*** (−1.01), ***PGD*** (−0.89) | *ACSF3* (−0.52), ***PGD*** (−0.92) |

Negative log₂FC corresponds to the reduced gene expression, positive log₂FC corresponds to the increased gene expression in ectopic cells. DEGs found in more than one cell type are marked in bold.
*DEGs* differentially expressed genes.
[a]Statistically significant values ($p_{adj}$ < 0.05).

**Table 3 | Statistically significant differentially expressed genes coding for proteins involved in the regulation of synthesis, conversion and metabolism of steroids in perivascular, endothelial, and stromal cell types of ectopic endometrium compared with eutopic endometrium**

| Related steroid | Gene | Cell type, log$_2$FC[a] ($p_{adj} < 0.05$) | | | Gene function |
|---|---|---|---|---|---|
| | | Perivascular | Stromal | Endothelial | |
| Progesterone | PGR | −1.94 | NS[b] | −4.66 | Progesterone receptor |
| | AKR1C1 | NS | 2.06 | 2.33 | Progesterone inactivation |
| | AKR1C2 | NS | 2.51 | NS | Progesterone inactivation |
| | HSD17B2 | NS | NS | −0.48 | Formation of active progesterone |
| Estrogen | ESR1 | −3.12 | −1.14 | −4.72 | Estrogen receptor alfa |
| | ESR2 | 1.84 | NS | NS | Estrogen receptor beta |
| | HSD17B8 | 1.00 | NS | NS | Interconversion of estrone and estradiol |
| | HSD17B2 | NS | NS | −0.48 | Estradiol conversion to estrone |
| Cholesterol | CYP11A1 | −3.98 | NS | NS | Steroid hormone precursor |
| Cortisol | HSD11B1 | NS | 2.55 | NS | Cortisone conversion to cortisol |
| | HSD11B2 | −1.60 | NS | NS | Cortisol conversion to cortisone |
| Androgen | AR | −0.93 | NS | NS | Androgen receptor |
| | SRD5A1 | −0.75 | NS | NS | Testosterone conversion to DHT |
| | SRD5A3 | −1.01 | NS | NS | Testosterone conversion to DHT |
| | HSD17B6 | NS | 5.53 | NS | Androgen catabolism |
| Ketones | HSD17B11 | NS | 1.16 | 1.18 | Ketone metabolism |

[a]Negative log$_2$FC corresponds to the reduced gene expression, positive log$_2$FC to the increased gene expression in ectopic cells.
[b]NS – statistically non-significant value ($p_{adj} > 0.05$).

binding. In endothelial cells, gene enrichment analysis identified extracellular structure and matrix organization, morphogenesis of a branching structure and macromolecule binding among the top biological processes and molecular functions.

## Discussion

In this single-cell transcriptomic study, we explored the gene expression from the perspective of cellular metabolism, steroidogenesis and cell cycle, in paired samples of EuE and EcE from women with endometriosis. To sustain cell survival, growth, and angiogenesis, endometriotic cells must be adaptable to adjust their metabolism based on cellular needs and environmental conditions. We found that three cell types of perivascular, stromal, and endothelial cells in EcE had different expression patterns of metabolic pathways compared to EuE (Fig. 2A–C and Table 1). We also observed a transcriptomic downregulation in progesterone signaling in these three cell types of EcE and an increased expression of genes associated with higher estrogen activity in perivascular cell population of EcE compared with EuE (Table 3). Moreover, in EcE, the proportions of perivascular, stromal and endothelial cells in G2/M and S phases were larger than in EuE (Fig. 2F).

The network of perivascular, stromal and endothelial cells plays a significant role in endometrial tissue growth. Stromal cell population comprises the most abundant cell type in endometrium and consists of multiple subpopulations[5,28] that actively interact with neighboring cells[29]. Perivascular cells are represented by two cell types, vascular smooth muscle cells and pericytes that reside in the area around large and small vessels, respectively, and interact with endothelial cells, promoting angiogenesis[30,31]. Pericytes were shown to act as mesenchymal progenitors and affect regenerative processes of endometrial stroma[32–35]. We found larger populations of perivascular, endothelial, and immune cells of EcE compared with EuE, whereas EuE was enriched with stromal cells (Fig. 1E), similar to the results reported by Zhu et al.[29]. Tan et al.[6] found endothelial cell enrichment in ectopic endometrium and reported a new subpopulation of perivascular cells characteristic to endometriosis, expressing gene markers MYH11 and STEAP4, which we also identified in both EcE and EuE (Supplementary Table 6).

Metabolic pathways are regulated by AMPK and HIF-1 signaling[22]. AMPK pathway is activated in response to energy stress by sensing increases in AMP:ATP and ADP:ATP ratios, leading to the activation of glycolysis and FA catabolism, an inhibition of macromolecular synthesis and cell cycle arrest[36,37]. The AMPK pathway, downregulated by steroid hormones, was shown to be involved in inflammation, metabolic regulation, and apoptotic processes in endometriosis[38]. HIF-1 signaling, a major regulator of oxygen homeostasis in cells, has also an impact on steroidogenesis and cell metabolism to sustain cell growth, angiogenesis and survival with a downstream activation of metabolic pathways in endometriotic lesions[18]. The first step of glycolytic way of energy production is glycolysis, during which glucose is converted into pyruvate with the production of 2 molecules of ATP. In oxidative metabolism, pyruvate is then converted to acetyl-CoA through the link reaction, which then enters the TCA cycle in the mitochondria. NADH and FADH (the products of glycolysis, FA oxidation and TCA cycle) are further utilized in electron transport chain consisting of OXPHOS enzyme complexes where ATPs are produced. FAs are one of the main sources of energy, and substrate for acetyl-CoA which is further used in oxidative metabolism. During oxidative metabolic process, a cell can produce around 30 molecules of ATP[39].

Some cells, in the presence of oxygen, switch from oxidative to glycolytic ATP synthesis, and pyruvate is converted into L-lactate or D-lactate by LDH enzymes, with regenerated NAD+ that is further re-used in glycolysis, thereby sustaining this reaction[39]. This metabolic switch is known as the Warburg effect, also referred as aerobic glycolysis, observed in cancer cells and in highly proliferative cells that aims to synthesize biomass and produce energy to maintain cell proliferation[23]. An altered energy metabolism toward aerobic glycolysis with inhibited oxidative respiration was observed in endometriosis studies on primates, and in vitro studies of endometriotic lesions, endometriotic stromal cells and mesothelial cells from women with endometriosis[19–21,40]. Inhibited AMPK signaling has been shown to lead to enhanced aerobic glycolysis in cancer cells[41]. A simultaneous activation of aerobic glycolysis and OXPHOS has been shown in cancer cells based on experimental[42,43] and transcriptome data analysis[44,45], and AMPK and HIF-1 signaling played a key regulatory role in sustaining

sufficient energy and macromolecular synthesis in these cells[44]. The role of L-lactate and D-lactate in glycolytic and oxidative metabolism was recently examined in cancer cells[46]. Remarkably, both forms of lactate were demonstrated to enter mitochondria, without being metabolized, and stimulate TCA cycle and OXPHOS.

Previous studies on cell metabolism in endometriosis have not examined perivascular cell niche and its potential role in the pathogenesis of endometriosis. In our study, we observed an overall HIF-1- and AMPK-mediated activation of glycolytic and oxidative metabolic systems in perivascular cells of EcE (Fig. 2B, C, Table 1), although we also identified downregulated activators of glycolysis, PFKFB3 and PFKFB4, and upregulated inhibitor of oxidative metabolism, PDK1. Interestingly, we found an activation of BRCA1 in perivascular cells of EcE, which was previously reported to inhibit glycolysis and induce oxidative metabolism in human breast cancer cell line[47]. We found an upregulation of genes that increase glucose uptake (TBC1D1 and GLUT4), key regulators of glycolysis (GCK, ENO2, PFKP, PGAM2, ALDOA and ALDOC), and overexpressed LDHA that mediates aerobic glycolysis. Moreover, there was a differential regulation of genes involved in production of glucogenic amino acid metabolism, like ASPA and GPT2. Ala, Asp and Glu glucogenic amino acids can be used to produce glycolysis intermediates in the energy depleted state of the cells. In parallel, there was an increased expression of PDHA1, PDHB and DLD, which encode enzymes from the pyruvate dehydrogenase (PDH) complex that links glycolysis and TCA cycle by converting pyruvate to acetyl-CoA. Additionally, in perivascular cells of EcE, we identified an upregulation of genes related to an increased FA uptake and transport to mitochondria (CD36, CPT1A and MLYCD), and activation of beta-oxidation of FAs with the production of acetyl-CoA (e.g. ACADL, CPT1A, CYP2U1). The downregulation of IDH2, shown to be involved in TCA cycle with glutamine as a substrate in cancer[48], could indicate a higher use of FAs and pyruvate over glutamine in TCA cycle. Furthermore, there was an upregulation of the key genes of TCA cycle (IDH3B, PCK1, SDHC, SDHD, SUCLA2 and SUCLG1), and the overexpression of 24% of genes encoding five OXPHOS multiprotein complexes. Collectively, these data might refer to the above-mentioned co-activation of aerobic glycolysis and oxidative metabolism in perivascular cells of EcE.

Within our analysis, we found less DEGs of glycolytic and oxidative metabolism in stromal cells than in perivascular cells of EcE. There was an upregulation of genes related to glucose uptake (TBC1D1 and GLUT4), key enzymes of glycolysis (ENO2 and PFKP), and downregulation of HK1. Similarly to perivascular cells, we observed an overexpression of LDH genes referring to aerobic glycolysis (LDHA and LDHD). The upregulation of PFKP, ENO2, and LDHA at protein and/or RNA levels have been reported in primary stromal cells from EcE compared to EuE and control endometrium[20]. Young et al.[21] demonstrated high levels of LDHA expression in EcE and an excess of lactate in peritoneal fluid from women with endometriosis. Recent findings have shown that in cancer cells, [13]C-labeled D-lactate, a substrate for LDHD, is not converted by cells for pyruvate or citrate. Instead, it stimulates pyruvate entry into mitochondria and OXPHOS, increasing cell proliferation and survival[46]. While the role of D-lactate has not been previously described in the context of endometriosis, the increased expression of LDHD in stromal cells of EcE may have a similar effect on stromal cell metabolism and proliferative activity via D-lactate. Apart from LDH family enzymes, we found an upregulation of DLD gene encoding enzyme from PDHA complex that associates glycolysis and TCA cycle. However, in vitro study of primary stromal cells from EcE has identified a low level of DLD compared to EuE and control endometrium[20]. In EcE, we also observed an increased expression of genes regulating FA transport to mitochondria and beta-oxidation (CPT1A, ACADL, ACADS, CYP2U1, etc.), and a downregulation of IDH2. In addition, there was an upregulation of two TCA key genes (SDHD and SUCLG1), and both up- and down-regulation of genes encoding OXPHOS complexes. Interestingly, four out of five downregulated genes of OXPHOS complexes (NDUFA6, COX5A, COX6C and UQCRQ), were also found to be less expressed at the protein level in cultured stromal cells from EcE compared with EuE in women with endometriosis and controls[20]. Among the five upregulated

genes that we found in OXPHOS pathway, two were downregulated at the protein level in cultured stromal cells from EcE[20]. Overall, genes related to both glycolytic and oxidative ways of energy production were activated in stromal cells of EcE, albeit to a lesser extent than in perivascular cells. We did not observe the distinct transcriptomic signature of the Warburg effect in the stromal cells of EcE as previously reported[20]. This discrepancy can be explained by the differences between ex vivo and in vitro experiments.

In the endothelial cell population of EcE, the overall gene expression pattern of metabolic pathways was similar to that in perivascular and stromal cells, but less altered. It can probably be explained by an analogous process of angiogenesis occurring in both EcE and EuE in the proliferative phase of menstrual cycle. In EcE compared to EuE, we found a differential regulation of glutathione peroxidases, reductases, transferases and other genes that are involved in glutathione synthesis and neutralization of oxygen reactive species that are mainly formed during oxidative metabolism[49]. AMPK signaling has an inhibitory effect on biosynthetic processes[50], which was observed in EcE through the downregulation of key regulators of gluconeogenesis (CREB5), FA and sterol synthesis (ACACB, SREBF1, LIPE, SCD, HMGCR), protein synthesis (RPTOR, EIF4EBP1, EEF2K), and glycogen synthesis (GYS1). Altogether, these results might suggest a more catabolic state of perivascular and stromal cells but not endothelial cells in EcE, with the co-activation of glycolytic and oxidative metabolism to sustain energy requirements.

Non-hormonal therapy of endometriosis has a potential to disrupt metabolic activity in the cells of EcE[51]. For example, McKinnon et al.[24] showed a high protein expression of glucose transporter GLUT4 in epithelial and stromal cells of EcE, suggesting GLUT4 role in glucose supply for lesions and proposing it as a target for therapy. Other targets for endometriosis management could be PDH and PDK1 that play key roles for oxidative and glycolytic metabolism. Their regulator dichloroacetate was studied by Horne et al.[26] on mice model of endometriosis, demonstrating reduction of endometriotic lesion size and lactate level in peritoneal microenvironment. A recent work on endometrial stromal and epithelial cell lines with LDHA knockdown has shown an increase in pro-apoptotic factors and reactive oxygen species, and a reduction in ATP production[25].

Steroid hormones play an important role in the regulation of adaptive metabolic response. The reduction of PGR (progesterone resistance), and an increase of E2 production as well as the prevention of E2 inactivation have been observed in endometriosis[52–56]. E2 regulates the transcription of genes involved in cell proliferation, survival and angiogenesis in EcE and other tissues, for example by activating PI3K/AKT and HIF-1 pathways[57–59]. E2 in endometrium is primarily bound to ERα, whereas in endometriosis ERβ mediates E2 effects and is overexpressed in both EuE and EcE, while ERα is downregulated in EcE[55,60,61]. ERβ was reported to be expressed in mitochondria and play a role in metabolic processes as well as in the prevention of apoptosis by regulating autophagy-related genes in endometriosis[7,62]. AKR1C1 and AKR1C2 are reductases that inactivate P4 in peripheral tissue, while 17β-HSD type 2 (encoded by HSD17B2) is known as an oxidative enzyme involved in the formation of an active metabolite of P4 (20α-DHP) and conversion of active E2 to inactive E1[52,63–65]. 17-beta-HSD 8 (encoded by HSD17B8) has been shown to exhibit oxidative and reductive activities in the interconversion of E2 to E1[66,67]. We examined the expression of genes involved in steroidogenesis in perivascular, stromal, and endothelial cells and observed a reduced expression of ESR1 in EcE. PGR was downregulated in perivascular and endothelial cells, and AKR1C1 and AKR1C2 were upregulated in stromal and endothelial cells. Perivascular cells exhibited a high expression of ESR2 and HSD17B8, while in endothelial cells, HSD17B2 was downregulated. Taken together, these data suggest an altered regulation of steroid hormones and downstream target genes of PGR, ERα and ERβ, and the activation of energy metabolic pathways in EcE (Fig. 3).

Cellular metabolism is a process tightly coupled with the cell cycle due to high demand for energy and substrates in a dividing cell. The stromal cells of EuE at the proliferative phase of menstrual cycle were found to be mostly in the G1 cell cycle phase of cell growth, characterized by active macromolecular synthesis[7,62]. McKinnon et al.[68] studied isolated endometrial stromal cells from women with and without endometriosis, and

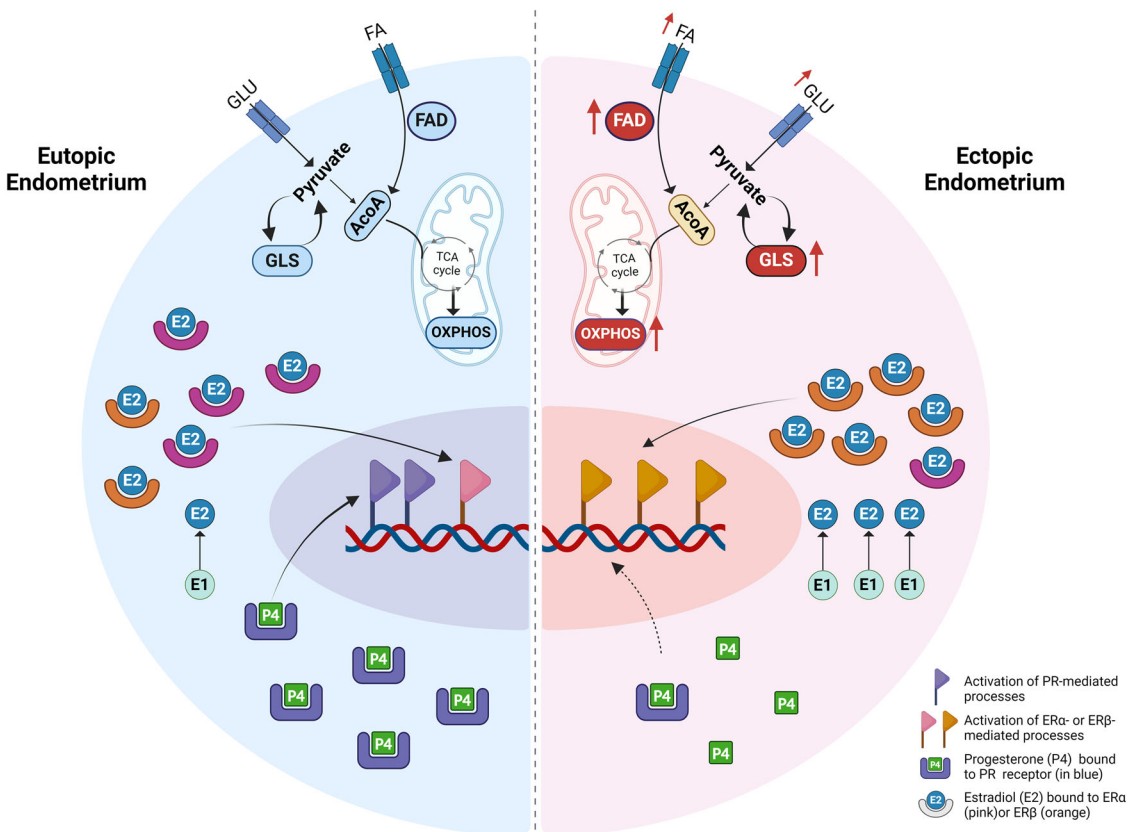

**Fig. 3 | A proposed model of possible interconnected mechanisms of steroid hormone regulation and altered cellular metabolism in perivascular cells of ectopic endometrium compared to eutopic endometrium.** In the ectopic endometrium (a half of a cell depicted on the right side) the following changes are observed compared to eutopic endometrium (on the left side): reduced production of progesterone receptor (PR, blue) contributing to progesterone (P4, green) resistance; decreased level of ERα and increased level of ERβ (in pink and orange, respectively) and increased conversion of estrone (E1, light blue) into estradiol (E2, blue), leading to a nuclear activation of aberrant set of estrogen target genes; upregulation of genes related to increased glucose and fatty acid (FA) uptake, activated FA degradation (FAD), and simultaneously increased OXPHOS and glycolysis (GLS) activities. FA fatty acids, GLU glucose, ACoA acetyl-CoA, nuclear gene expression is indicated with flags, black arrows refer to sequence of the processes, red arrows refer to the activation of processes, central dashed line separates the halves of the perivascular cells of eutopic and ectopic endometrium. Created with BioRender.com.

found that a large portion of the cells was in G1 phase (70%) and only 15% of the cells were in G2M or S phases. We identified more than 60% of perivascular, stromal and endothelial cells of EuE in G1 phase (Fig. 2F), which correlates with the above-mentioned reports. In EcE, we observed larger proportions of cells expressing genes from S and G2/M phases.

Our scRNA study has some limitations. Firstly, the sample size was limited to four women with endometriosis from the proliferative phase of menstrual cycle. Despite this, we identified the major cell populations in both EuE and EcE, revealing their steroidogenic and metabolic cell type heterogeneity. A sample-wise UMAP (Supplementary Fig. 3) demonstrated a similar cell population pattern within each group (EuE or EcE). Our differential expression analyses which encompass a pseudobulk approach, compared biological samples rather than individual cells to ensure each patient's contribution by considering the variability between the patients. The latter was visually confirmed by the representative plots of DEGs in metabolic pathways (Supplementary Figs. 5–7), which demonstrated transcriptomic similarity of metabolic activity between samples within each group (EuE vs EcE). The GSVA analysis further confirmed the metabolic pathway activity for each patient (Supplementary Fig. 4). Moreover, both the samples of EuE and EcE were derived from the same women. These paired samples make the results more robust. Secondly, as our study included only the samples from women with endometriosis, we conducted the metabolic activity analysis on EuE from women with endometriosis vs controls using external dataset[27]. The results showed virtually no difference in the activity of metabolic pathways and steroidogenesis between the two types of EuE samples (Supplementary Table 4). Thirdly, the study focused

exclusively on transcriptomic data analysis with no functional validation of the results. However, previous studies on human cancer have reported that transcriptomic changes in metabolism might refer to and predict metabolic activity in cells[69,70]. Further studies may include larger and more diverse patients' populations for ethnicity and endometriosis stage, to explore interpatient and interpopulation variability of metabolic reprogramming in endometriotic lesions with functional analysis. In addition, we observed a small population of epithelial cells in both EuE and EcE, most likely due to the cell loss during the tissue dissociation procedure. Some previous studies also reported smaller populations of epithelial cells in EcE compared to EuE of women with endometriosis and controls[6,29].

## Conclusions

Within this study we gained an insight into metabolic heterogeneity, steroidogenesis and cellular proportions in cell cycle phases in EcE and EuE. We found changes in the regulation of progesterone and estrogen signaling pathways in perivascular, stromal and endothelial cells of EcE compared with EuE, which might have a direct effect on cellular metabolism and contribute to cellular proliferation and angiogenesis. The metabolic pathways were differentially regulated in perivascular, stromal and to a less extent in endothelial cell populations, and metabolic pathway activity alterations between EcE and EuE were the highest for AMPK, HIF-1, glutathione metabolism, OXPHOS, and glycolysis/gluconeogenesis. Remarkably, we identified a transcriptomic co-activation of glycolysis and OXPHOS in perivascular and stromal cells of EcE compared to EuE, possibly to meet the increased energy demands of proliferating cells.

Perivascular cells, known to contribute to the restoration of endometrial stroma as well as angiogenesis, may be an important population within EcE to sustain the growth and survival of endometriotic lesions. We observed altered metabolic activity and steroidogenesis predominantly in perivascular cells of EcE, supporting our hypothesis. These findings may facilitate further development of alternative strategies of non-hormonal treatment targeting metabolic pathways in endometriosis.

## Materials and methods

The study was approved by the Research Ethics Committee of the University of Tartu, Estonia (No 333/T-6), and written informed consent was obtained from all participants. All ethical regulations relevant to human research participants were followed.

### Patient selection and sample processing

The paired EuE and EcE samples were collected from four women with endometriosis undergoing laparoscopic surgery at the Tartu University Hospital (Tartu, Estonia). All women were in their follicular phase of the menstrual cycle (days 7–11), 33 ± 6.4 years old (mean ± standard deviation), with normal body mass index of 21 ± 1.8 kg/m$^2$ and had not received any hormonal treatments for at least 3 months prior to the time of laparoscopy. According to the revised American Society for Reproductive Medicine classification system[71], three women had minimal-mild endometriosis and one woman had moderate-severe endometriosis. During the laparoscopic surgery, peritoneal lesions were removed, and endometrial samples were collected using an endometrial suction catheter (Pipelle, Laboratoire CCD, France). All specimens were cut into two pieces and one portion was immediately fixed in 10% formalin for tissue histological evaluation, and the other portion stored in cryopreservation medium containing 1× Dulbecco's Modified Eagle's Medium (DMEM, Thermo Fisher Scientific), 30% fetal bovine serum (FBS, Biowest, Riverside, MO, USA) and 7.5% Dimethyl Sulfoxide Hybri-Max (Sigma-Aldrich). Tissue samples in cryopreservation medium were placed into Nalgene Cryo 1°C 'Mr Frosty' Freezing Container (Thermo Scientific) and deposited into a −80 °C freezer overnight. The frozen biopsies were then stored in liquid nitrogen until further use. The haematoxylin and eosin staining and IHC analysis for Ki67 marker in EuE and EcE were performed in Tartu University Hospital Pathology Department following standard protocols. The presence of endometriosis-specific morphological features in the lesions was confirmed by a pathologist and the images were prepared using Aperio Image Scope software v12.4.6.5003 (Aperio Technologies Inc.) (Fig. 1B and Supplementary Fig. 15).

### Tissue dissociation for scRNA-seq

Cryopreserved tissues (50–80 mg) were warmed in a water bath at 37 °C until thawed. Immediately after thawing, the biopsy content was washed twice in a 15 ml falcon tube with a 5–7 ml pre-warmed DMEM medium (phenol red free, charcoal-stripped FBS 10% + antibiotics/antifungal, Corning), to remove cryoprotectant and excess blood. The tissues were dissociated using an enzymatic cocktail of 2.5 mg/ml collagenase I (Sigma), 0.25 mg/ml DNase (AppliChem GmbH) and 10 mg/ml Dispase II (Gibco) in 10 ml of the DMEM medium. After shaking the tubes vigorously, further dissociation was continued for around 1 hour in an incubator at 37 °C by placing the tubes in the horizontal position on a rotating shaker. The resulting suspension of the tissues was left unagitated for 1–2 min in a vertical position to settle the undigested tissue fragments by gravity. Any undigested tissue material was re-digested enzymatically in an incubator for another 15 min. The single-cell suspension was filtered through a 30-μm cell strainer to remove debris and cells were pelleted by centrifugation at 300 × $g$ for 5 min. Red blood cells were removed by the treatment with ACK lysis buffer (Gibco), following the manufacturer's instructions. Pelleted single cells were then resuspended in 500 μl of DMEM medium and filtered again through a 30 μm cell strainer to collect the suspension in a 5% BSA-coated 1.5 ml microfuge tube. The cells were counted using Trypan blue chemistry on the automated counter (Corning) to determine the concentration and viability of the cell suspension. Live cells were enriched using MACS dead cell removal kit (Miltenyi Biotec) following the manufacturer's instructions. The enriched cell suspensions having viability >90% were washed with 0.04% BSA (Sigma) in Dulbecco's DPBS (1×, Gibco) on ice for three times to remove ambient RNAs. The cells were resuspended in 0.04% BSA in DPBS to achieve a final concentration of 700–1200 cells/μl, as recommended by the manufacturer's protocol (10x Genomics).

### Chromium 10x single cell capturing, library generation and sequencing

Single cell RNA libraries were constructed using 10x Chromium Next GEM Single Cell 3' reagent v3.1 kit (Dual index, 10x Genomics, CG000315 Rev C) comprised of single cell 3' gel bead kit, library construction kit and chip G kit as per the manufacturer's instructions (10x Genomics). Single-cell suspensions, single-cell 3' gel beads, and master mix for reverse transcription reagents were loaded onto a 10x Chromium microfluidic chip for a targeted cell recovery of 3000 cells per sample. The samples were processed in a 10x Chromium controller to generate Gel Beads-in-emulsion (GEM). Further steps, including cell lysis, first-strand cDNA synthesis, amplification, and purification, were carried out as per the manufacturer's instructions to produce barcoded full-length cDNA. Library preparation was carried out simultaneously for all the samples using the library construction kit according to manufacturer's instructions to avoid the batch effect. Quality control of cDNA and single-cell libraries were analyzed using Agilent 4150 tape station (Agilent Technologies). Dual indexed single-cell libraries were pooled, and pair-end sequenced targeting 35,000 reads per cell using NovaSeq PE150 (Illumina).

### Quality control, normalization, doublet cell removal, sample integration, and cell-type clustering

The 10x Genomics scRNA-seq data was aligned with refdata-gex-GRCh38-2020-A reference genome downloaded from the 10x website, and count matrices for each sample were created using the Cell Ranger (v7.0.0) software.

The 10x Genomics scRNA-seq data were analyzed with the CRAN package Seurat. In the data processing procedure, we retained cells with a UMI count of more than 500 and cells with the number of genes more than 200. High-complexity cell types (> 0.80) were kept, and cells with more than 15% of mitochondrial reads were filtered out. Genes with zero counts were removed by keeping only genes that were expressed within 3 or more cells. Following which, each sample was normalized, and the cell cycle score was assigned by the CellCycleScoring function based on the expression of G2/M and S phase markers collected from Tirosh et. al., according to the Seurat package manual[72]. The analysis was based on the expression of 98 cell cycle genes corresponding to cell cycle phases: S phase of DNA synthesis (43 genes), G2/M phase of checkpoint and cell division (combined G2 and M phases, 55 genes) and G1 phase of cell growth (cells expressing none of the 98 cell cycle genes). SCTransform method of normalizing, estimating the variance of the raw filtered data, and identifying the most variable genes was applied to regress out the source of variation by mitochondrial expression, and cell cycle phase. Next, the SCTransformed samples were subjected to Principal Component Analysis (PCA), and a minimum number ($n = 40$) of PCs were identified. Based on the resulting PCs, we ran UMAP clustering, FindNeighbors, and FindClusters with 0.1 resolution steps from Seurat. Next, the pre-processed data was submitted to the DoubletFinder[73] R package with default parameters. We then removed the identified cell doublets from each sample before integrating all eight samples. 5000 highly variable shared features were used to integrate the samples to identify shared subpopulations across EuE and EcE. Canonical correlation analysis was performed to identify shared sources of variation between the groups. Briefly, canonical correlation analysis identifies the greatest sources of variation in the data, but only if it is shared or conserved across the conditions/groups using the 5000 variable features. In the second step of integration, the mutual nearest neighbors were identified, and incorrect anchors (cells) were filtered out to integrate the samples across conditions. After integration, to visualize the integrated data, we used dimensionality reduction techniques, PCA, and UMAP. FindClusters with resolutions ranging from 0.2 to 1.4

from Seurat were used to find clusters in the integrated data. The UMAP technique was used to visualize the cell clusters in merged and individual samples for each group (EuE and EcE). This integrated dataset with identified cell clusters was used for further downstream analysis.

For the analysis of external dataset[27] (GEO accession code GSE214411) including EuE from women with endometriosis ($N = 3$) and EuE from women without endometriosis (controls, $N = 3$) in the proliferative phase of menstrual cycle, we followed the same steps of the analysis from quality control to cell clustering as for our dataset, and filtered out cells with ribosomal reads more than 15%.

## Cell cluster annotations

We used two approaches to identify and label the major cell types in the integrated dataset. First, the FindAllMarkers function from the Seurat package was used to find the differentially expressed genes from each identified cell class with default parameters. Next, we prepared a cell marker list collected from literature[6,7,28,74,75], Cell Marker database[76], Panglao database[77], and single-cell RNA database from the Human protein atlas to validate the identified cell markers (Supplementary Table 6). To shortlist the marker genes, we applied $\log_2 FC > 1$, adjusted $p$-value < 0.05, PCT.1 ≥ 0.7 (higher expression of a marker in a specific cluster) and PCT.2 ≤ 0.3 (lower expression of the same marker in other clusters). Statistical analysis of cell proportions, comparison of total cell numbers and cell numbers by cell cycle phases between EcE and EuE was performed using two-sided Fisher's exact test. For the analysis of external dataset, it was subjected to Bioconductor package SingleR (v 2.6.0) and we performed unbiased cell type annotations by using our dataset as a reference.

## Pseudobulk differential expression and KEGG pathway analyses

We incorporated pseudobulk analysis, which aggregates scRNA-seq counts across individual samples, allowing us to infer population-level gene expression patterns. To ensure robust and biologically relevant comparisons, differential expression analysis was conducted between different conditions (e.g., EuE vs. EcE) using four distinct samples from each group. Thus, our strategy ensured that our comparisons are based on diverse biological samples rather than individual cells. Pseudobulk differential expression analysis between the cell populations of EcE and EuE and for each cell cycle phase was carried out using DESeq2 (v.1.40.1) R package[78]. The gene enrichment and the KEGG pathway enrichment analyses of differentially regulated genes (adjusted $p$-value < 0.05, $\log_2 FC \leq -0.40$ and ≥0.40) in each cell type were performed using clusterProfiler (v.4.8.1)[79,80] R package. Pseudobulk differential expression analysis between the cell clusters of EuE from controls (normal endometrium) and EuE from women with endometriosis was carried out using DESeq2 (v.1.40.1) R package, in the same way as for our dataset. Venn diagram of overlapping KEGG pathways was made using online tool Venny 2.0[81].

## Metabolic and steroidogenesis pathways analysis

We selected 12 pathways related to cellular energy metabolism from the KEGG pathway database available online (Supplementary Table 7). For each pathway gene list, we extracted the differential expressions in EcE vs EuE from the output of DESeq2 pseudobulk differential expression analysis as mentioned in the previous step. Similarly, the genes involved in steroidogenesis were checked (Supplementary Table 8). The same strategy was used to analyze metabolic pathways and steroidogenic genes in the eutopic and control endometria from external dataset[27]. The bar plot representing the numbers of DEGs in 12 metabolic pathways across 9 major cell types and the total number of genes in a given pathway (Fig. 2A) was made using GraphPad Prism 6 software.

## The single-cell pathway analysis

We used the SCPA method[82] to rank the 12 selected metabolic pathways between EcE and EuE, in each cell type. SCPA analysis assesses the changes in the multivariate distribution of a pathway and not just pathway enrichment. We used Q-value (multivariate distribution) measurement as a primary statistic, as suggested by the authors of the SCPA method.

## Gene set variation analysis

We applied Gene Set Variation Analysis (GSVA), a non-parametric and unsupervised method to obtain the selected 12 pathway gene set overrepresentation in each patient to understand the patient variation on metabolic pathway level. We used Bioconductor package GSVA[83] (v. 1.52.2).

## Transcription factor analysis

The transcription factor (TF) activity inference from our scRNA-seq dataset was carried out using decoupleR (v. 2.10.0) package[84]. We performed the TF activity on the genes from 12 metabolic pathways including all 8 samples. To check the TF activity separately in EuE and EcE samples, we extracted count matrix for EuE and EcE from the integrated scRNA-seq data and mapped TF gene to the target gene.

## Statistics and reproducibility

The statistical analyses of data and sample size were described in the respective subsections of Methods and Results and included in the respective Figure legends. To compare cell proportions between EcE ($N = 4$) and EuE ($N = 4$) and cell cluster proportions between EcE and EuE for each cell cycle phase groups, we utilized two-sided Fisher's exact test. The statistical significance was determined by $p_{adj}$ values less than 0.05. R and GraphPad Prism 9.0.0 (GraphPad, San Diego, CA) were used for data analyses and visualization.

## Reporting summary

Further information on research design is available in the Nature Portfolio Reporting Summary linked to this article.

## Data availability

The raw transcriptomic data (Supplementary Data 3) is available online. Supplementary Data 3 was deposited at the GEO repository and is available under the GEO accession number GSE247695. All scripts and details on processing steps were deposited in GitHub and are available at: https://github.com/ankita16lawarde/scRNA-Seq_endometriosis.

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

## Acknowledgements

We acknowledge the use of BioRender.com to create the Figs. 1A, 2E and 3. This research was supported by the European Union's Horizon 2020 research and innovation MATER program under grant agreement No. 813707 (M.Sar.), Estonian Research Council grants PRG1076 (A.S.), MOBJD1056 (A.P.), Horizon 2020 innovation grant ERIN, grant no. EU952516 (A.S.), Enterprise Estonia grant no EU48695 (M.P. and M.Saa.), Horizon Europe (NESTOR, grant no. GA101120075, M.P., A.S.) and MSCA-RISE-2020 project TRENDO grant no 101008193 (A.S.).

## Author contributions

A.S., M.P., M.Sar., and A.L. designed the study. P.S. collected samples and clinical data from patients, analyzed clinical data. M.Sar., A.P., and M.Saa. performed wet-lab experiments. A.L. and V.M. performed the bioinformatic analysis. M.Sar., A.P., M.Saa., and M.P. performed the analysis and inter-pretation of bioinformatic results. M.Sar., A.L., A.P. and M.Saa. wrote the manuscript. M.Sar., A.L. and A.P. visualized the graphical data. M.P., M.Saa., A.S., K.G.D., P.G.L.L., T.K. and A.T. contributed to the interpretation of study results and discussions, revised and edited the manuscript. All authors read and approved the final version of the manuscript.

## Funding

## Competing interests

The authors declare no competing interests.
