## [Peer Review File · Communications Biology]

Reviewers' comments:

Reviewer #1 (Remarks to the Author):

Sarsenova et al's manuscript titled 'Endometriotic lesions exhibit distinct metabolic signature compared to paired eutopic endometrium at the single-cell level' used single-cell RNA-sequencing to reveal which cell types of endometriotic lesions are metabolically altered and promote their growth. They found that metabolic pathways were differentially regulated in perivascular and stromal cells, especially in AMPK signaling, HIF-1 signaling, glutathione metabolism, oxidative phosphorylation, and glycolysis/gluconeogenesis. The author provides the insight about the critical role of metabolic reprogramming in maintaining cell growth and survival of endometriotic lesions. This study provides valuable resource and important findings for the corresponding field. Before acceptance, some problems should be addressed. Importantly, the readability should be improved for general readers, and some additional bioinformatics analysis should be added.

Major points

1. Pathological sections (for example, H&E staining) should be provided to display the pathological characteristics of EcE and EuE samples.
2. In this manuscript, the author detected several DEGs in indicated cell types, and these upregulated and downregulated DEGs were associated with diverse metabolic pathways. However, for a given metabolic pathway and a given cell type, some genes were upregulated in EcE compared with EuE, but some other genes were downregulated, which had been described in this manuscript. The descriptive details were relatively complex and easily confused the readers. Maybe the author uses more condensed language to describe their findings. Moreover, the SCENIC analysis may be helpful to infer the potential core regulators in the corresponding metabolic pathway. The author also should plot the gene expression level of the representative genes, for example, using boxplot.
3. The author only focused on the metabolic pathways in this manuscript. However, other enrichment analysis should be performed based on DEGs in different cell types, and simply describe the results of the enrichment analysis.
4. The author detected the altered proliferation and apoptosis, and if conditions permit, I suggest to use the immunofluorescence staining to verify these findings rather than the bioinformatics analysis.

Minor points

1. Figure 1A, the authors utilized the t-SNE plot in the schema on the reduction analysis, but in this study, there is no t-SNE plot being indicated. Please check whether there is something wrong in this plot.
2. Lines 98 and 101, the statistics method should be indicated. For example, two-tailed Student's t-test.
3. Lines 116-128, the visualization bar plot could be utilized to display the different percentage across cell types.
4. Lines 142, 147 and others, did the author indicate 'all cell types together' instead of 'all cell clusters together'?
5. Line 165, did the author indicate 'up- and down-regulated DEGs' instead of 'up- and down-

regulated genes'? Additionally, the author should describe the method to calculate DEGs? Lines 666-671, the authors describe the DEG analysis in pseudobulk samples, but it's confused where and why this analysis was performed.

6. Figure 2C, the author should cite the corresponding reference.

7. Line 186, for the readability, the author should clearly indicate 'all three cell types'.

8. Line 380, what is the definition of 'altered genes'? In the whole manuscript, the author should use the unified proper noun.

Reviewer #2 (Remarks to the Author):

The authors present a manuscript where they have performed single cell RNA sequencing on paired eutopic and peritoneal lesions from women with endometriosis. They have concentrated their focus on the evaluation of metabolic pathways and demonstrate that the most abundant and profound changes were evident perivascular, stromal and endothelial cells.

Although the biological concept is interesting (but not novel) and clearly important in the pathogenesis of endometriosis, the current manuscript is limited in terms of depth of information presented and lacks validation and biological mechanism.

The main pathways that are over represented and the genes involved in those are shared, this information is useful for others in the field but the work does not represent a significant advance in knowledge for the field.

Major comments:

i) the authors could and should interrogate the single cell data in more depth. What is the contribution per patient, are all the pathways equally over represented in each patient? It is important to visualise patient variation. Could metabolic pathways be a mechanism to stratify patients? N number is small, can this analysis be applied to other datasets?

ii) the authors include a large section in the results discussing the over represented pathways and genes that are up and down regulated. It would be more appropriate for the discussion of these results to be in the discussion section.

iii) it would be useful to compare to control eutopic endometrium. Could integrate data from other datasets.. are metabolic pathways over represented in eutopic endometrium from patients with endometriosis compared to control?

iv) I'd have really liked to see to deeper dive into what all this means with some mechanistic work.

Minor comments:

i) In figure 1B the label 'cycling stroma' has moved under macrophages

Dear Reviewers,

Thank you for your valuable feedback, we have incorporated all the changes to our revised manuscript according to your comments and tried to make sure that we addressed each point. We believe that the additional analyses and in-text corrections improved and refined the content and flow of our revised manuscript.

Reviewer	Major/minor	Comment	Response
Reviewer 1	Major	1. Pathological sections (for example, H&E staining) should be provided to display the pathological characteristics of EcE and EuE samples.	1. We have included the images (Fig. 1B and Suppl. Fig. 15) of H&E staining of both eutopic (EuE) and ectopic (EcE) endometrial samples from two patients, from whom we obtained the paired samples for this scRNA-seq study (page 7, lines 111 - 113). Generally, the lesions are very small pieces of endometriotic tissue and due to their limited size, it was not feasible to divide them for both staining and for scRNA-seq experiment, which requires a large number of cells. Due to this reason, unfortunately, for the other two patients we had no EcE material left for H&E staining.
		2. In this manuscript, the author detected several DEGs in indicated cell types, and these upregulated and downregulated DEGs were associated with diverse metabolic pathways. However, for a given metabolic pathway and a given cell type, some genes were upregulated in EcE compared with EuE, but some other genes were downregulated, which had been described in this manuscript. 1) The descriptive details were relatively complex and easily confused the readers. Maybe the author uses more condensed language to describe their findings. 2) Moreover, the SCENIC analysis may be helpful to infer the potential core regulators in the corresponding metabolic pathway. 3) The author also should plot the	2. 1) We have revised and refined the text to describe (in the Results section) and discuss (in the Discussion section) the results more clearly and concisely for readers aiming to avoid confusing and complex descriptions of the findings. We hope the text is now easier to follow. 2) Due to computational challenges with SCENIC, we opted for decoupleR to infer transcription factor (TF) interactions in our scRNA-seq analysis. DecoupleR, a Bioconductor package, uses the Univariate Linear Model (ULM) method to fit a linear model for each sample and TF, predicting gene expression based on TF-gene interaction weights. Therefore, our analysis using DecoupleR provides a comprehensive regulatory landscape, capturing extensive TF interactions with genes from metabolic pathways and ensuring reliable insights into regulatory mechanisms. Our results are detailed in the manuscript on page 10, lines 177 – 185, Suppl. Table 4 and visualized in Fig. 2D. 3) We added boxplots (Suppl. Fig. 5 - 7) of representative metabolic DEGs from four groups (Regulatory pathways, Glycolytic metabolism, Oxidative metabolism and

		gene expression level of the representative genes, for example, using boxplot.	Biosynthetic pathways, from the Table 2 in the manuscript) and indicated them in the text (page 11, lines 200 - 201).
		3. The author only focused on the metabolic pathways in this manuscript. However, other enrichment analysis should be performed based on DEGs in different cell types, and simply describe the results of the enrichment analysis.	3. In the Results section of the manuscript, we describe the results of the enrichment analysis (page 18, lines 367 - 374) of perivascular, stromal and endothelial cell types under the subsection "Differential expression and pathway analyses, revealed altered proliferation, apoptosis, migration and angiogenesis processes in EcE". The GO plots with the top 10 biological processes (BP), molecular function (MF) and cell compartment (CC) (Suppl. Fig. 14) have been added to the manuscript.
		4. The author detected the altered proliferation and apoptosis, and if conditions permit, I suggest to use the immunofluorescence staining to verify these findings rather than the bioinformatics analysis.	4. We conducted the immunostaining for proliferation marker Ki67 of both eutopic (EuE) and ectopic (EcE) endometrial samples from three patients (Fig. 2G and Suppl. Fig. 12). The results are described in the Results section of the manuscript under "Differential expression and pathway analyses, revealed altered proliferation, apoptosis, migration and angiogenesis processes in EcE" subsection (page 17, lines 353 - 357), and in the Materials and Methods section (page 26, lines 610 - 614).
Minor		1. Figure 1A, the authors utilized the t-SNE plot in the schema on the reduction analysis, but in this study, there is no t-SNE plot being indicated. Please check whether there is something wrong in this plot.	1. We apologize for the confusion. This figure was created in Biorender.com using a template of data visualization plot for scRNA-seq cluster graph. We have now corrected the y/x axes titles in the representative figure (Fig. 1A, page 6) from tSNE to UMAP and added the names of cell types to the legends.
		2. Lines 98 and 101, the statistics method should be indicated. For example, two-tailed Student's t-test.	2. We have now indicated in the text the statistical method used to compare the cell proportions for each cell type between EuE and EcE (page 4, lines 99 - 101).
		3. Lines 116-128, the visualization bar plot could be utilized to display the different percentage across cell types.	3. The bar plot representing 12 metabolic pathways (the number of DEGs between EcE and EuE to the total number of genes in a given pathway) across all the cell populations has been added to the manuscript as a Fig. 2A (page 9, lines 156 – 158, page 19, lines 389 – 391, page 30, lines 729 – 731).

		4. Lines 142, 147 and others, did the author indicate ‘all cell types together’ instead of ‘all cell clusters together’?	4. The corrections (the word "clusters" was replaced by the words "types" or "populations") have been made throughout the whole text, except for the parts describing bioinformatic analysis and results of cell clustering.
		5. Line 165, did the author indicate ‘up- and down-regulated DEGs’ instead of ‘up- and down-regulated genes? Additionally, the author should describe the method to calculated DEGs? Lines 666-671, the authors describe the DEG analysis in pseudobulk samples, but it’s confused where and why this analysis was performed.	5. The respective corrections have been made specifying the "DEGs" instead of "genes" (page 10, line 186). The method of calculating DEGs is now described in more detail (page 29, lines 713 - 721) in the Materials and Methods section ("Pseudobulk differential expression and KEGG pathway analyses" subsection). In short, pseudobulk analysis was used to compare scRNA-seq counts from each sample to analyze gene expression pattern based on the individual samples rather than individual cells.
		6. Figure 2C, the author should cite the corresponding reference.	6. We apologize for this confusion, the corresponding figure (currently as a Fig. 2E) has been created by the first author using BioRender.com tool.
		7. Line 186, for the readability, the author should clearly indicate ‘all three cell types’.	7. The respective corrections in the text have been made throughout the manuscript.
		8. Line 380, what is the definition of ‘altered genes’? In the whole manuscript, the author should use the unified proper noun.	8. The "altered genes" are now replaced with "up- and down-regulated DEGs" term (page 17, line 346) that has been used across the manuscript.
Reviewer 2	Major	i) the authors could and should interrogate the single cell data in more depth. What is the contribution per patient, are all the pathways equally over represented in each patient? It is important to visualise patient variation. Could metabolic pathways be a mechanism to stratify patients? N number is small, can this analysis be applied to other datasets?	i) We have added the UMAP figure (Suppl. Fig. 3), representing the cell clustering in each sample within each group (EuE and EcE). The UMAP shows the cell cluster similarity across biological samples from the same sample group (EuE or EcE) indicating the homogeneity of the sample groups, mentioned in the manuscript text (page 4, lines 90 – 92, page 24, lines 551 - 552). Additionally, we have attached the boxplots of representative metabolic DEGs from four groups of metabolic pathways showing the range of the level of expression across the samples (Suppl. Fig. 5 – 7, page 11, lines 200 - 201). To identify and visualize patient variation, we also performed GSVA analysis, which assesses the activity of each metabolic pathway in each sample. The results showed a similar

			metabolic pattern across the samples of the same sample group for each of the three cell types (perivascular, stromal and endothelial cells), and no specific sample contributed solely to the metabolic activity results of each sample group (Suppl. Fig. 4, page 10, lines 174 - 177). Unfortunately, we could not use this analysis to see if patients could be stratified according to metabolic processes. The sample size of our study was relatively small, and the data of other scRNA-seq studies on endometriosis with publicly available raw transcriptomic data, are extremely heterogeneous. Those samples were from different phases of the menstrual cycle, some of the enrolled patients were using hormonal medications, or having other gynecological conditions. Thus, much larger study group would be needed to perform such an analysis using samples matching our study groups, such as samples from the proliferative phase of the menstrual cycle, peritoneal endometriosis, no hormonal treatment and no other gynecological conditions present.
		ii) the authors include a large section in the results discussing the over represented pathways and genes that are up and down regulated. It would be more appropriate for the discussion of these results to be in the discussion section.	ii) We have revised and re-organized the text in the Results section (in particular, the Metabolic activity parts) and moved the respective parts to the Discussion section to improve the clarity and the flow of each section of the manuscript.
		iii) it would be useful to compare to control eutopic endometrium. Could integrate data from other datasets. Are metabolic pathways over represented in eutopic endometrium from patients with endometriosis compared to control?	iii) We performed the analysis of metabolic activity in EuE (proliferative phase of menstrual cycle, endometriosis stages III and IV) from women with (N=3) and without (N=3) endometriosis from the external dataset (Huang et al. 2023, GEO accession code GSE214411). The results of this analysis are added to the Results (Suppl. Table 6, Suppl. Fig. 8 – 9, in the text of subsection “Metabolic activity is similar in EuE from women with endometriosis vs controls”, pages 15 – 16, lines 291 – 298, 320 - 322) and are mentioned in the Discussion part on the study

			limitations (page 24, lines 560 - 563) and in the Methods (pages 28, 29, lines 689 – 693, 717 – 720, 727 - 729).
		iv) I'd have really liked to see to deeper dive into what all this means with some mechanistic work.	iv) We have discussed the main results in more detail with the reference to the published studies that included experimental validation of the findings. In terms of the results from perivascular and endothelial cell populations to date there is no published data on cell metabolism of perivascular niche in the context of physiology or pathology. However, we tried to integrate the reported data on metabolism of stromal cells that were studied in vitro (primary stromal cell cultures) and we discussed the similarities and differences between our findings and possible reasons for the discrepancy. The main challenges with comparing our findings to the published data from endometriosis studies are: the focus of our study on cell metabolism, which was not highlighted previously in the context of endometriosis; the different types of tissues used (some studies on cell metabolism in endometriosis were focusing on peritoneum-derived mesothelial cells), menstrual cycle phases used in the reported studies, and the type of the experimental studies (e.g. in vitro , ex vivo , etc.)
	Minor	i) In figure 1B the label 'cycling stroma' has moved under macrophages	i) As cycling stromal cluster in EcE was represented only by few cells (due to this low cell count the cluster name was displaced under the cluster of macrophages), we have decided to remove this cluster from EcE on the UMAP (Fig. 1C). We mention in the title of the Fig. 1C: "The cycling stromal cell cluster in EcE was represented by few cells and due to the low cell count and wide distribution of the dots representing this cluster, the cluster name has been omitted from the figure to avoid confusion with localization of the cluster."
Other changes			Additionally, we shortened the Abstract (155 words) and re-arranged the positions of figures 1 – 3 in the text. Small Supplementary Tables (1-3; 6 - 10) and Supplementary Figures are

			added as a single pdf file "Supplementary_files". Large data tables (4 & 5) are uploaded separately in Excel format.
--	--	--	--